# Angiogenin Levels and Carotid Intima-Media Thickness in Patients with Type 1 Diabetes and Metabolic Syndrome

**DOI:** 10.3390/biomedicines11092591

**Published:** 2023-09-21

**Authors:** Jolanta Neubauer-Geryk, Melanie Wielicka, Grzegorz M. Kozera, Leszek Bieniaszewski

**Affiliations:** 1Clinical Physiology Unit, Medical Simulation Centre, Medical University of Gdańsk, 80-210 Gdansk, Poland; melanie.wielicka@gmail.com (M.W.); gkozera@gumed.edu.pl (G.M.K.); lbien@gumed.edu.pl (L.B.); 2Department of Pediatrics, Northwestern University Feinberg School of Medicine, Division of Neonatology, Ann Robert H. Lurie Children’s Hospital of Chicago, Chicago, IL 60611, USA

**Keywords:** angiogenin, carotid intima-media thickness, atherosclerosis, type 1 diabetes mellitus, metabolic syndrome

## Abstract

It is well documented that in patients with type 1 diabetes (DM1), decreased levels of angiogenin are associated with the development of overt nephropathy. However, little is known about angiogenin levels and subclinical macrovascular organ damage in patients with DM1 and concomitant metabolic syndrome (MS). Therefore, we analyzed the relationship between angiogenin levels and carotid intima-media thickness (cIMT) in DM1 patients with and without MS. We found that angiogenin concentration was significantly lower in DM1 patients compared to controls, while the cIMT measurements were comparable. Exclusion of patients with MS, patients with hypertension, undergoing treatment, or cigarette smokers did not change these findings. Of note, when comparing the subgroups of DM1 patients with and without MS, there was no significant difference between angiogenin levels. However, we did note a significant difference in these levels after the exclusion of smokers. The comparison of cIMT in these subgroups showed a significant difference between the study subgroups. This difference was no longer observed when the age of the patients was taken into account. In summary, it can be concluded that metabolic syndrome in patients with type 1 diabetes does not appear to impact angiogenin levels or cIMT.

## 1. Introduction

Although the mortality risk related to type 1 diabetes has declined by 29% over the past 10 years, it is still estimated to be between two and eight times higher than that of the general population [1,2]. Thus, the ESC (European Society of Cardiology) guidelines recommend management of cardiovascular risk factors in patients with type 1 diabetes, particularly those over the age of 40 or with evidence of microvascular complications [3]. It is estimated that in 2021, there will be about 8.4 million people living with diabetes; 18% of patients were under the age of 20, 64% aged 20–59, and 19% aged 60 and over. It is estimated that one million new cases will be diagnosed in 2023, with a median age of onset of 39 years. For a 10-year-old child diagnosed with type 1 diabetes in 2021, the life expectancy was estimated to range from 13 years in low-income countries to 65 years in high-income countries [1].

Angiogenin is a pro-angiogenic cytokine that increases endothelial cell activity and enhances their proliferation and migration in the early stages of angiogenesis [4]. It is a factor with anti-inflammatory [5] and neuroprotective [4,5,6] properties. It influences the formation of prostacyclin and NO by activating NOS, which acts as a vasodilator [6]. Angiogenin plays an important role in the process of angiogenesis in both physiological and pathological states, such as wound healing, tissue repair processes, or tumor growth [7]. Reduced serum angiogenin levels in patients with diabetes may be responsible for impaired angiogenesis, especially in those with long-standing disease [8,9]. Chiarelli et al. showed angiogenin levels in children with type 1 diabetes increased as early as puberty. They demonstrated that the severity of microangiopathic complications in those patients was associated with a significant increase in angiogenin levels, while long-term strict glycemic control resulted in a decrease in angiogenin levels [10].

Hyperglycemia is associated with increased protein glycation, vasoconstriction, decreased nitric oxide levels, inflammation, and increased reactive oxygen species formation. All of these processes lead to endothelial dysfunction, which drives the pathogenesis of several disease processes. Both morphological and mechanical tests can be used to objectively evaluate endothelial function. These methods allow for the assessment of the thickness of the intima-media complex and the examination of arterial compliance. Arterial blood flow changes with endothelial dysfunction, which can be exacerbated by the presence of cardiovascular risk factors [11]. Endothelial dysfunction not only plays a key role in the pathogenesis of atherosclerosis through the associated loss of elasticity and reduction in compliance but is also closely associated with the development of retinopathy and nephropathy [12].

While these processes are the result of endothelial dysfunction, they also contribute to it, intensifying vascular damage and leading to the earlier onset of both micro- and macroangiopathic vascular complications. Carotid intima-media thickness (cIMT) is widely used by clinicians and researchers as a marker of subclinical macrovascular damage, representing the progression of atherosclerotic disease. Atherosclerosis is a manifestation of arterial aging and is further augmented by the presence of hypertension, lipid disorders, diabetes, smoking, or inflammatory processes [13].

Recent results from a meta-analysis carried out by Meimei et al. in August 2023 suggest that there is an association between ultrasonographic parameters of the carotid vessels and both microvascular and macrovascular complications of diabetes. Therefore, Meimei et al. proposed that in patients with diabetes, carotid ultrasound may be a valuable tool for predicting patient outcomes [14]. Abnormalities in parameters such as carotid intimacy, media thickness, pulse wave velocity [15], and central amplification index have the potential to show early signs of vascular pathology. The researchers [16] found that there was an independent association between CIMT and cerebral microbleeds in young individuals with type 1 diabetes despite no clinical signs of neurological disease, indicating a link between subclinical systemic atherosclerosis and early microvascular pathology in the brain. In a meta-analysis, Wang et al. [17] showed that cIMT, endothelium-dependent flow-mediated dilation (FMD), carotid-femoral pulse wave velocity, and glyceryl trinitrate-mediated dilatation are significantly different between patients with type 1 diabetes and control patients. Their results are in line with the current evidence for increased CV risk in type 1 diabetic patients and stress the need for identifying new markers of subclinical atherosclerosis. Another review indicated that arm FMD is lower and cIMT is higher in obese adults. Consequently, changes in arm FMD and cIMT may be important surrogate markers appropriate for use in clinical trials [18]. It should be noted that cIMT is more widely used in everyday clinical practice. The Epidemiology of Diabetes Interventions and Complications (EDIC) or Diabetes Control and Complications Trial (DCCT) found an association between cIMT and subsequent coronary events in patients with type 1 diabetes. However, this association was no longer found after adjusting for traditional cardiovascular risk factors [19]. 

Atherosclerosis progresses over a number of years in the absence of symptoms. Therefore, early identification of patients who are at risk of developing symptomatic disease offers avenues for primary prevention. Previous studies have shown that various interventions that reduce cIMT progression are associated with a significant reduction in CVD (cardiovascular disease) risk. These findings further support the use of cIMT as a marker of increased risk for atherosclerosis, potentially helping identify individuals at risk for early development of advanced disease [20]. 

Similar to the general population, a combination of poor nutrition and a sedentary lifestyle in patients with type 1 diabetes promotes weight gain and can lead to visceral obesity, insulin resistance, and hypertension. Metabolic syndrome refers to a cluster of risk factors widely accepted for identifying patients at high cardiovascular risk. The DCCT/EDIC trial [21] revealed that metabolic syndrome occurs in about 34% of patients with type 1 diabetes. 

The term metabolic syndrome (MS) was introduced to emphasize the cumulative effect of several risk factors. Various studies have shown an increased risk of CVD in MS, regardless of the MS criteria used [22,23,24,25,26]. Mottillo found a 2-fold increased risk of CVD, CVD mortality, and stroke and a 1.5-fold increased risk of mortality from any cause associated with MS in the largest systematic review and meta-analysis (n = 951,083) [27]. However, more recent studies have shown that the associations between MS and CVD outcomes are either null or weak [28,29]. The RIVANA [30] study, conducted in Mediterranean populations, found that MS was independently associated with incident CVD, CVD-related mortality, and all-cause mortality. However, its association with myocardial infarction or stroke could not be established. Each MS component was also independently associated with all study outcomes, with similar strength to MS alone. This study showed that the risk of cardiovascular events, with the exception of myocardial infarction, increased along with the increasing number of MS components. However, there may be differences in the risk of cardiovascular events depending on the combination of individual criteria.

Whether defined by the ATP III or the IDF criteria, metabolic syndrome consists of five metabolic abnormalities and indicates a substantial lifetime risk of cardiovascular disease. However, it is not a reliable indicator of 10-year cardiovascular risk. A global risk assessment, such as the Framingham score, is still necessary for the determination of 10-year risk [31].

We believe that central obesity and its metabolic consequences play a substantial role in the progression of type 2 diabetes and hypertension, as well as the exacerbation of any preexisting conditions. In fact, overweight patients with type 1 diabetes experience hyperinsulinemia and insulin resistance, leading to the development of double diabetes. Merger et al. [32] performed a study of 31,119 people with type 1 diabetes, searching for patients with MS and examining them for late vascular complications. The study revealed that 25.5% of this population had both T1DM and MS. Furthermore, those with dual diabetes showed an increased prevalence of macrovascular comorbidities, including coronary heart disease (8.0% compared to 3.0% without MS), stroke (3.6% compared to 1.6%), and diabetic foot syndrome (5.5% compared to 2.1%). Microangiopathic complications were found to be twice as frequent as in the group without MS (retinopathy: 32.4% versus 21.7%, nephropathy: 28.3% versus 17.8%). Vascular complications were more common in the double diabetes group, regardless of glycemic control. The team therefore concluded that it is important to identify patients at risk of double diabetes and prevent them from developing MS.

It has been demonstrated that angiogenin may play a role in atherosclerotic plaque formation [33]. Therefore, we decided to study the influence of metabolic syndrome on the relationship between angiogenin concentration and intima-media thickness (cIMT) in patients with type 1 diabetes. Our report is dedicated to the hypothesis that the presence of metabolic syndrome is associated with increased angiogenin concentration and cIMT. 

## 2. Materials and Methods

### 2.1. The Study Design and Population

The study group consisted of 56 patients with type 1 diabetes (31 women and 26 men, age: 39 ± 6.6 years) and 38 control subjects (31 women and 25 men, age: 37.1 ± 6.6 years) (Table 1). We included patients with a minimum of 4.5 years of diabetes duration. The principal consideration for enrolling in the study was to sign an informed consent to enter the study. The study did not include diabetic patients who met the following exclusion criteria: renal insufficiency, uncontrolled diabetes or ketoacidosis, chronic respiratory disease, history of cardiovascular events, history of clinical manifestations of disorders of cerebral circulation, and/or focal symptoms of central nervous system damage confirmed by neurological examination. In addition, patients with severe hypoglycemia within 30 days prior to the examination were excluded from the cohort. The factor that excluded healthy individuals, or those with diabetes, as well as diabetic patients, from the study was the presence of systemic diseases such as rheumatoid arthritis or psoriasis. People were included if they had a diagnosed medical history of both hypothyroidism and hyperthyroidism and were euthyroid at the time they qualified for the study. Euthyroidism was confirmed by testing TSH and fT4 hormone levels in the month prior to the study. In the absence of euthyroidism, these tests were performed as part of the biochemical tests. Pregnancy and alcohol abuse were also reasons for exclusion in both groups. Patients were recruited from the Diabetic Outpatient Clinic, while controls were recruited from coworkers and their families. The study was conducted according to the guidelines of the Declaration of Helsinki and approved by the Medical Ethics Committee of the Medical University of Gdansk (NKEBN/335/2008, NKEBN/335-60/2009, and NKEBN/204/2010), and each participant gave informed consent upon entry into the study. The study protocol consisted of taking a medical history concerning existing disorders, comorbidities, and cigarette smoking, neurological examination [34], fundoscopy [35], and laboratory testing. Serum angiogenin levels were measured utilizing the enzyme-linked immunosorbent assay (ELISA) Quantkine kit (R D Systems, Minneapolis, MN, USA) according to the manufacturer’s instructions. The CIMT assessment was measured offline using a semi-automatic method (Carotid Measure System) [36]. To maintain an appropriate blood glucose range during testing, blood glucose was checked by the patient using their own blood glucose meter. Body temperature was also closely monitored. A stable temperature was maintained during the examinations.

### 2.2. Metabolic Syndrome Criteria

Diabetic patients were classified into subgroups with or without metabolic syndrome (Table 1) according to the 2005 International Diabetes Federation (IDF) criteria [37]. Based on these criteria, a diagnosis of metabolic syndrome requires central obesity defined by waist circumference (women > 80, while men > 94 cm) or BMI > 30 kg/m^2^. In addition, 2 of 4 criteria must be present for diagnosis, among which are: TG ≥ 150 mg/dL or treatment of hypertriglyceridemia; HDL < 40 mg/dL (men); < 50 mg/dL (women) or treatment of low HDL; blood pressure: systolic ≥ 130 mmHg; diastolic ≥ 85 mmHg or treatment; fasting blood glucose ≥ 100 mg/dL or previously diagnosed treatment; fasting blood glucose ≥ 100 mg/dL or previously diagnosed diabetes mellitus [37].

### 2.3. Statistical Analysis

All statistical analysis of the data obtained was performed using STATISTICA version 13.1 statistical software from StatSoft Inc., Tulsa, OK, USA; license CSM GUMed JPZP5077539317AR-H.

The distribution of the variables was assessed using the Shapiro-Wilk test. In the absence of a normal distribution of the study variables, their values were compared using non-parametric tests—the Mann-Whitney U test. A comparison of variables with a normal distribution, presented as a mean value (SD—standard deviation), was carried out using parametric tests—the Student’s *t*-test. Relationships between variables were tested using Spearman’s rank correlation and the Chi^2^ test, with Yates correction when appropriate. The chi-square test was used to compare gender proportions and the prevalence of hypertension, microangiopathy, contraceptive use, medication use, and hypoglycemic episodes. Intergroup comparisons for related continuous variables were made using a general linear model with Fisher’s exact test for post-hoc analysis, if necessary. A significance level of *p* < 0.05 was considered statistically significant.

## 3. Results

There was no significant difference between age, gender, BMI, smoking, statin, or oral contraceptive use between the patients with diabetes and the control group (Table 1). The total cholesterol level and all its fractions, as well as blood pressure, did not significantly differ between the two groups. Hypertension and active therapy with cardiac medications were more prevalent in the study group. HbA1c, CRP, and urine albumin/creatinine ratio levels were also significantly higher in diabetic patients (Table 2).

Angiogenin levels were significantly higher in the control group compared to the study group, even after excluding patients with metabolic syndrome, those treated with medications, and active smokers. The difference was no longer significant following the exclusion of patients with microangiopathy. CIMT did not differ between patients with diabetes and controls, even after excluding patients with metabolic syndrome, those treated with medications, smokers, or patients with microangiopathy (Table 2).

There were no significant differences in angiogenin levels between men and women in the studied groups (Table 1). CIMT in the control group did not differ in the subgroups separated by gender, while cIMT in men with diabetes was significantly thicker compared to that in women (0.53 (0.39–0.74) vs. 0.51 (0.40–0.63) mm, *p* = 0.01).

The study group subjects were classified into patients with metabolic syndrome (with MS) and without metabolic syndrome (without MS). Subgroup analysis revealed that patients with MS were significantly older and had a later onset of diabetes compared to patients without MS. Diabetes duration was comparable between both subgroups. There were significantly more men in the subgroup with MS compared with the subgroup without MS. The subgroups did not differ in the presence of angiopathy, the number of hypoglycemic episodes, HbA1c levels, the number of smokers, or the number of patients taking oral contraceptives or other medications.

Blood pressure, CRP, and urine albumin/creatinine ratio were also comparable between the groups. Triglyceride levels were higher in the subgroup with MS, but the remaining cholesterol fractions did not differ between subgroups.

Serum angiogenin levels were comparable between the subgroups even after the exclusion of patients taking medications and those with microangiopathy (Figure 1). However, they were significantly higher in the subgroup with metabolic syndrome after exclusion of smokers [456.5 (230.6–580.6) vs. 3416 (190.4–500.4) ng/mL, *p* = 0.03] (Table 2). CIMT differed between the diabetic patients with and those without metabolic syndrome (Figure 1), even after excluding patients treated with medications, smokers, and those with microangiopathy (Table 2). In all diabetic patients, we noted a significant positive correlation between angiogenin levels and age (r = 0.33, *p* = 0.01), insulin intake (r = 0.31, *p* = 0.02), creatinine level (r = 0.45, *p* > 0.001), and package-years (r = 0.27, *p* = 0.04). In this group, significant positive correlations were shown between cIMT and age (r = 0.55, *p* > 0.001), as well as age of diabetes onset (r = 0.28, *p* = 0.04), BMI (r = 0.46, *p* > 0.001), and creatinine level (r = 0.41, *p* = 0.002). The calculated power of the test yielded 0.828.

After adjusting for gender and age, the significant difference in cIMT values between subgroups of diabetic patients was no longer observed (Table 3). After adjusting for gender and age, insulin intake, creatinine level, and smoking habit, significant differences in angiogenin levels between subgroups of diabetic patients were still not present.

## 4. Discussion

Little is known about angiogenin levels as a marker of subclinical macrovascular complications in adult patients with type 1 diabetes and concomitant metabolic syndrome. In our study, we analyzed the effect of metabolic syndrome on the relationship between angiogenin levels and cIMT in patients with type 1 diabetes. Our present study revealed that angiogenin levels were significantly higher in the control group compared to the diabetic group, even after excluding patients with metabolic syndrome, those taking medications, and smokers. Concurrently, when analyzing the subgroups of type 1 diabetic patients, a significant difference in angiogenin concentration was found only after the exclusion of smokers.

In 2009, researchers agreed that abdominal obesity should not be a prerequisite for the diagnosis of metabolic syndrome but rather only one of the five criteria. Currently, the presence of three out of the five criteria is required for diagnosis [38].

It should be mentioned that although metabolic syndrome is predominantly identified in overweight individuals, it can sometimes occur in lean people. An important finding from the Osadnik study [39] was that the presence of MS in lean patients carries the same adverse long-term prognosis as in obese patients.

In our study, we used the IDF criteria, which identified patients with numerous cardiovascular risk factors beyond type 1 diabetes. All study participants with MS met the waist criterion, and four of them met the BMI criterion. Of note, some of the patients did not meet the full diagnostic criteria for MS despite central obesity.

We did not use the ATP III criteria in our study as we felt they were not sufficiently restrictive. According to the IDF, 35.7% of diabetic patients in our study had MS, while according to the ATP III, only 28.6% would meet the full criteria. The interesting new criteria proposed by the Polish researchers are based on the presence of central obesity (waist 88/102 cm or BMI > 30 kg/m^2^), carbohydrate disorders, hypertension, and lipid disorders expressed as non-HDL cholesterol levels or hypolipemic treatment. Only 12.5% of our patients met the Polish criteria for metabolic syndrome [40].

To our knowledge, there are no published reports on the relationship of angiogenin levels with carotid intima-media thickness in adult patients with type 1 diabetes with metabolic syndrome.

Despite the fact that our study did not demonstrate differences in cIMT between type 1 diabetic patients and controls, we found a significant difference in cIMT when comparing diabetic patients with and without metabolic syndrome (Table 2). This difference remained significant even after the exclusion of subjects with microangiopathy, medication intake, or smoking (Table 1). CIMT in patients with diabetes and metabolic syndrome is thicker compared to diabetics without metabolic syndrome.

Published reports on angiogenin levels in type 1 diabetics and their association with metabolic status present inconsistent conclusions. These papers seem to take into account only certain components and not the full spectrum of metabolic syndrome. Chiarelli et al. report that serum angiogenin levels were significantly higher in children as well as in adolescents and young adults with diabetes compared to controls [10]. Additionally, they found that angiogenin concentration had a positive correlation with HbA1c levels and concluded that significant, long-term improvement in glycemic control is associated with a reduced level of angiogenin. In turn, Malamitsi [41] found higher angiogenin levels in young diabetic patients compared to healthy controls and reported that angiogenin concentration did not correlate with glycemic control.

Our data analysis of the relationship between angiogenin and metabolic status differs from prior studies. We showed that the angiogenin level in patients with type 1 diabetes was significantly lower compared to the control group. It should be noted, however, that the patients we studied were older (38.7 ± 6.3 (25–53) yo.) (Table 1) than the patients in Chiarelli’s (age 3–29.7 yo.) [10] and Malamitsi-Puchner‘s study (14.3 ± 3.6 (5.5–21) yo.) [41]. They also had a longer disease duration. We did not find any significant correlation between angiogenin concentration and HbA1c, BMI, or lipid levels.

The relationship between angiogenin concentration and long-term type 2 diabetes control assessed by HbA1c levels has been studied by several researchers [8,9,42]. Siebert et al. [8] indicated that in patients with type 2 diabetes, the level of angiogenin is higher in those with poor glycemic control (HbA1c ≥ 7.0) than in patients with HbA1c < 7. When comparing patients with diabetes to controls, they found that angiogenin concentrations were significantly lower in patients with type 2 diabetes [9]. Höbaus et al. showed no differences in angiogenin levels between the group with type 2 diabetes and the control group. Moreover, angiogenin levels did not correlate with glycemic control or lipid levels, with the exception of triglycerides [43].

Marek-Trzonkowska et al., in their study, showed lower angiogenin concentrations in hypertensive patients in comparison to the control group [44]. Moreover, they showed that patient BMI as well as TG and LDL fractions were not associated with angiogenin levels.

Angiogenin in diabetes is a paradoxical substance, as summarized in the reviews of the literature on angiogenin by Yui [45] and Fadini et al. [46]. Angiogenin, among other factors, contributes to vascular complications in diabetes. On the other hand, its deficit leads to impaired vascular regeneration [45]. Abnormal angiogenesis in diabetes leads to both capillary rarefaction (bone marrow, myocardium, and peripheral nerves) [47,48] and neovascularization (proliferative retinopathy, atherosclerotic plaque) [49,50,51].

A study of pediatric diabetic patients by Chiarelli et al. [10] demonstrated that in this age group, angiogenin levels increase before puberty and are associated with the development of retinopathy and nephropathy. Our previous study of middle-aged patients with type 1 diabetes demonstrated significantly higher angiogenin levels in those who have already developed nephropathy when compared to those who have not [52]. These findings seem to be in line with Chiarelli et al. in that they suggest angiogenin levels may be a marker for the presence of microvascular complications [10].

Some studies have reported increased angiogenin expression in atherosclerotic lesions, suggesting its potential role in promoting neovascularization and plaque progression [33,42]. On the other hand, recently, Su’s et al. research has shown the protective role of endothelial angiogenin in atherosclerosis. Angiogenin expression was reduced in human and mouse atherosclerotic lesions. These findings suggest that angiogenin may play an important role in atherosclerosis therapy [53]. Kręcki revealed that patients with three-vessel coronary artery disease had higher angiogenin levels than the control group [33]. Burgmann et al. demonstrated that the level of angiogenin was increased in patients with peripheral artery disease in Fontaine stage IV and can be considered a marker of vascular disease progression [42]. However, this conclusion was not confirmed in a recent study by Höbaus [43].

Intima-media thickness in type 1 diabetes has been studied as a marker of early subclinical atherosclerosis [54]. Moreover, it has been shown that cIMT is the most useful biological indicator of vascular aging in both sexes [55]. In DCCT, intima-media thickness was thicker in patients with MS in comparison to those without MS [56]. The authors of the SEARCH CVD study found that BMI was the only identified risk factor that predicted cIMT at this age. It is very possible that high lipid levels, hypertension, smoking, and HbA1c are associated with cIMT, but only after longer terms of exposure [57]. In our study, we did not show differences in cIMT between the diabetic patients and the controls. However, there was a significant difference in cIMT when comparing the subgroups with and without MS (Table 2).

With regards to patient gender, the Baltimore Longitudinal Study on Aging revealed thicker cIMT in men compared to women in the general population. Our present analysis showed that cIMT was significantly thicker in men compared to women within the diabetic group. However, this correlation was not found in the control group. Of note, the ages of men and women in both groups were comparable [58]. It is a well-known fact that cIMT correlates with age, thickening on average by 0.1 mm per 10 years [55]. In our study, cIMT also strongly correlates with the age of subjects, both diabetics and controls. After adjusting for the age of the studied subjects, the difference between cIMT values in the diabetic patient subgroups with and without MS was no longer significant (Table 3).

The limitation of the present study is the relatively small number of subjects. The imbalance between the number of participants in each subgroup may be a limitation of our study, which could potentially influence the power of the study. Our results require further studies with larger cohorts of patients to validate the results obtained. Another limitation of our study was the use of cIMT as the sole marker of atherosclerosis and vascular function. The use of additional methods would provide valuable additional information. For instance, FMD could help understand endothelial dysfunction in diabetic patients with metabolic syndrome. This demonstrates the potential for future studies in this area [59]. Additionally, discussion of the results we have obtained was somewhat difficult due to the lack of other reports on angiogenin and cIMT in patients with metabolic syndrome and type 1 diabetes.

## 5. Conclusions

Angiogenin plays a significant role in the pathogenesis of endothelial dysfunction. Based on our data, abnormal angiogenesis in patients with type 1 diabetes may be related to lower angiogenin levels. However, the presence of concomitant metabolic syndrome in these patients was noted in angiogenin levels.

CIMT, which is widely used as a marker of macroangiopathy in patients with type 1 diabetes, differed significantly between subgroups of diabetic patients with or without metabolic syndrome. This difference was no longer observed when patient age was taken into account.

In summary, it can be concluded that metabolic syndrome in patients with type 1 diabetes does not appear to impact angiogenin levels or cIMT.

## Figures and Tables

**Figure 1 biomedicines-11-02591-f001:**
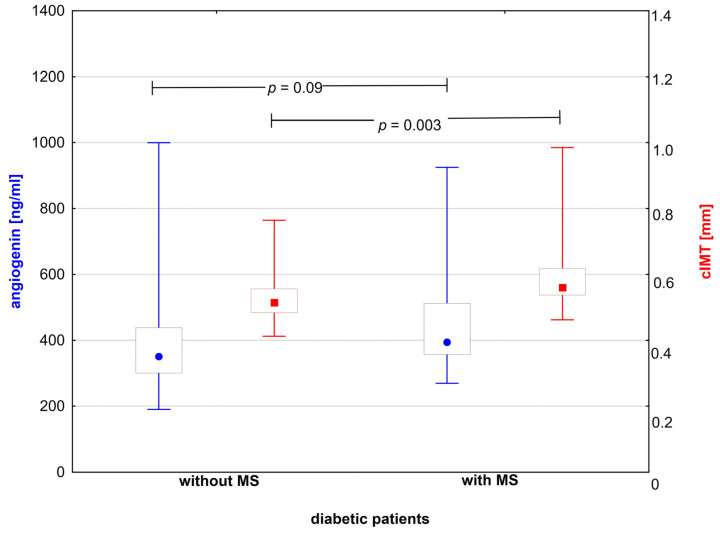
cIMT and angiogenin concentration in subgroups of diabetic patients with or without metabolic syndrome. Red squares and blue dots indicate median for respective variable.

**Table 1 biomedicines-11-02591-t001:** Characteristics of diabetic patients and matched control subjects.

	Control Group	Diabetic Patients	*p* for Between-Group Comparison	Diabetic Patient Subgroups According to IDF 2005		*p* for Between-Subgroup Comparison
Characteristics	n = 34	n = 56		Without MS n = 36	With MS n = 20	
	Mean ± SD and Median (Range)			Mean ± SD and Median (Range)		
Males, n (%)	14 (41.2%)	25 (44.6)	0.75	12 (33.3)	13 (65)	0.02
Age, years	37.1 ± 6.6	38.7 ± 6.3	0.31	36.6 ± 5.1	40.1 ± 7	0.03
	(26.3–51.3)	(25–53)		(25–48)	(33–54)	
Onset of diabetes [age]	n.a.	19.2 ± 7.1	n.a.	17 ± 6.3	20.5 ± 9	0.048
		(3–39)		(3–29)	(3–39)	
Duration of diabetes [years]	n.a.	19.5 ± 6.6	n.a.	18.7 ± 7.1	19.54 ± 6	0.85
		(4.5–35.3)		(4.5–35.3)	(12–35)	
Hypertension, n (%)	0	10 (17.9)	0.02	3 (8.3)	7 (35)	0.03
ACEI/ARB/ B-blockers, n (%)	0	10 (17.9)	0.02	5 (13.9)	5 (25)	0.5
Smokers, n (%)	3 (8.8)	11 (19.6)	0.28	9 (25)	2 (10)	0.32
Package-years, n	0 (0–20)	0 (0–20)	0.4	0 (0–20)	0 (0–19)	0.42
Hormonal contraception, n (%)	3 (8.8)	3 (5.4)	0.8	2 (5.6)	1 (5)	0.6
Microangiopathy, n (%)	n.a.	34 (61.7)	n.a.	20 (55.6)	14 (70)	0.29
Statin treatment, n (%)	0	6 (10.7)	0.12	3 (8.3)	3 (15)	0.75
Episodes of severe hypoglicaemia [N/last year]	n.a.	0 (0–10)	n.a.	0 (0–5)	1 (0–10)	0.07
Episodes of mild hypoglicaemia [N/last month]	n.a.	4 (0–30)	n.a.	4 (0–30)	4 (0–30)	0.85
BMI [kg/m^2^]	23.6	24.5	0.37	23.1	28.5	<0.001
	(20.1–33.3)	(19–27.3)		(19–27.2)	(5.1–37.3)	
Systolic blood pressure [mmHg]	130	133	0.1	130	139	0.18
	(94–189)	(83–180)		(83–180)	(103–168)	
Diastolic blood pressure [mmHg]	70	66	0.47	65	70	0.56
	(45–113)	(45–120)		(45–120)	(48–86)	

Legend: ACEI—angiotensin-converting enzyme inhibitor; ARB—angiotensin receptor blocker; MS—metabolic syndrome; IDF—International Diabetes Federation; *p* < 0.05 was considered statistically significant.

**Table 2 biomedicines-11-02591-t002:** Characteristics of diabetic patients and matched control subjects’ laboratory results and cIMT.

	Control Group	Diabetic Patients	*p* for Between-Group comparison	Diabetic Patient Subgroups According to IDF 2005		*p* for Between-Subgroup Comparison
Characteristics	n = 34	n = 56		Without MS n = 36	With MS n = 20	
	Mean ± SD and Median (Range)			Mean ± SD and Median (Range)		
HbA1c [%]	5.5	7.61	<0.001	7.5	8.2	0.11
	(4.5–6.1)	(6–10)		(6–10)	(6.5–9.8)	
Insulin dose units/24 h	n.a.	53	n.a.	48	69	<0.001
		(24–109)		(24–94)	(40.9–109)	
Insulin dose units/kg	n.a.	0.71	n.a.	0.71	0.79	0.69
		(0.36–1.16)		(0.36–1.16)	(0.46–1.13)	
C-reactive protein [mg/L]	0.7	1.59	<0.001	1.2	2.65	0.01
	(0.2–9)	(0.3–9)		(0.3–7.3)	(0.4–9)	
Serum creatinine [mg/dL]	0.75	0.79	0.2	0.79	0.79	0.48
	(0.7–1.1)	(0.6–1.2)		(0.6–1.0)	(0.65–1.17)	
Urine albumin/creatinine ratio [μg alb./mg creat.]	6.87	11.7	0.002	10.4	12.6	0.22
	(1.8–25.3)	(1.2–1016.3)		(1.2–835.8)	(5.9–1016.3)	
Total cholesterol [mg/dL]	194	185.5	0.35	176.5	190	0.12
	(128–278)	(143–302)		(143–302)	(157–236)	
Cholesterol HDL [mg/dL]	50.5	58	0.07	60.5	57	0.81
	(33–90)	(33–115)		(36–111)	(33–115)	
Cholesterol LDL [mg/dL]	126	112.5	0.09	109.5	117	0.35
	(79–185)	(49–214)		(49–214)	(60–169)	
Triglicerides [mg/dL]	80	65	0.32	60	85.5	0.01
	(35–223)	(38–299)		(38–299)	(46–191)	
Cholesterol non-HDL [mg/dL]	140	126	0.09	120.5	133.5	0.14
	(89–220)	(57–256)		(57–256)	(75–187)	
Serum angiogenin [ng/mL]	455.1	372.3	0.03	351	394.4	0.09
	(230.6–580.6)	(190.4–999.8)		(190.4–999.8)	(270–925)	
Serum angiogenin [ng/mL] after the exclusion of patients with IDF criteria	453.6	351	0.02	n.a.	n.a.	n.a.
	(230.6–580.6)	(190.4–999.8)				
Coefficients of variation for serum angiogenin [%]	20.52	41.6				
Serum angiogenin [ng/mL] after the exclusion of patients with microangiopathy	455.1	370.3	0.13	343.3	440.5	0.94
	(230.6–580.6)	(199.8–999.8)		(199.8–999.8)	(272.4–924.8)	
Serum angiogenin [ng/mL] after the exclusion of patients with hypertension and/or treatment with ACEI/ARB/B-blockers	455.1	369.2	0.02	341.6	440.5	0.5
	(230.6–580.6)	(190.4–999.8)		(190.4–999.8)	(272.4–924.8)	
Serum angiogenin [ng/mL] after the exclusion of smokers	456.5	365.4	0.003	341.6	394.4	0.02
	(230.6–580.6)	(190.4–924.8)		(190.4–500.4)	(269.6–924.8)	
cIMT [mm]	0.52	0.54	0.06	0.5	0.56	0.003
	(0.4–0.7)	(0.4–1.0)		(0.4–0.8)	(0.46–0.99)	
cIMT [mm] after the exclusion of patients with IDF criteria	0.51	0.52	0.55	n.a.	n.a.	n.a.
	(0.4–0.6)	(0.4–0.8)				
cIMT [mm] after the exclusion of patients with microangiopathy	0.52	0.53	0.62	0.5	0.55	0.04
	(0.39–0.66)	(0.41–0.66)		(0.4–0.7)	(0.52–0.63)	
cIMT [mm] after the exclusion of patients with hypertension and/or treatment with ACEI/ARB/B-blockers	0.52	0.53	0.10	0.51	0.56	0.001
	(0.4–0.7)	(0.4–1.0)		(0.41–0.77)	(0.52–0.99)	
cIMT [mm] after the exclusion of smokers	0.52	0.54	0.09	0.51	0.56	0.006
	(0.4–0.7)	(0.4–1.0)		(0.4–0.8)	(0.46–0.99)	

Legend: HbA1c—glycated hemoglobin; ACEI—angiotensin-converting enzyme inhibitor; ARB—angiotensin receptor blocker; cIMT—intima-media thickness; MS—metabolic syndrome; IDF—International Diabetes Federation; *p* < 0.05 was considered statistically significant.

**Table 3 biomedicines-11-02591-t003:** Characteristics of cIMT and angiogenin concentration in subgroups of diabetic patients with or without metabolic syndrome.

Parameter	Adjusted for	Diabetic Patient Subgroups According to IDF 2005		*p* for Between-Subgroup Comparison
		Without MS n = 36	With MS n = 20	
cIMT [mm]		0.5 (0.4–0.8)	0.56 (0.46–0.99)	0.003
	gender			0.051
	age			0.25
	age of DM onset			0.047
	creatinine			0.03
Serum angiogenin [ng/mL]		351 (190.4–999.8)	394.4 (270–925)	0.09
	gender			0.26
	age			0.67
	insulin/24 h			0.89
	smoking			0.30
	creatinine			0.60

Legend: cIMT—intima-media thickness; DM—diabetes mellitus; MS—metabolic syndrome; IDF—International Diabetes Federation; *p* < 0.05 was considered statistically significant.

## Data Availability

The research data can be requested from the first author.

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
