# Peer review of "Angiogenin Levels and Carotid Intima-Media Thickness in Patients with Type 1 Diabetes and Metabolic Syndrome"

_biomedicines, 2023, doi:10.3390/biomedicines11092591_

Round 1
Reviewer 1 Report
-
No comments.
Author Response
We kindly thank the Reviewer for a positive opinion about our study.
Reviewer 2 Report
It is a quite interesting paper. The aim of the study was to identify the relationship between the angiogenin levels and carotid intima-media thickness (cIMT) in diabetes 1. patients with and without metabolic syndrome. The Authors concluded that the presence of metabolic syndrome in patients with type 1 diabetes was not reflected in both angiogenin levels and cIMT.
The Authors have presented sufficient data. The appropriate tables and figures have been provided. The article is easy to read and logically structured. The methods are adequately described. The authors used appropriate statistical methods. The conclusions are consistent with the presented evidence and arguments.
There are some comments in the reviewer's opinion that should be taken under consideration by the Authors:
1. Please add the inter- and within-test coefficients of variation (CV) of Angiogenin
2. Please add more limitations to your study: In future studies, it will be worth estimating the correlations between angiogenin and endothelial dysfunction estimated by flow-mediated dilation. Brachial flow-mediated dilation (FMD) is a physiologic measure and Carotid IMT is an anatomic structural measure of subclinical atherosclerosis [PMID: 36141513; PMID: 27765534]; too small groups, etc.
3. In the introduction section, please add more information about metabolic syndrome and its influence on CV risk.
Author Response
Reviewer #2:
Comments and Suggestions for Authors
It is a quite interesting paper. The aim of the study was to identify the relationship between the angiogenin
levels and carotid intima-media thickness (cIMT) in diabetes 1. patients with and without metabolic
syndrome. The Authors concluded that the presence of metabolic syndrome in patients with type 1
diabetes was not reflected in both angiogenin levels and cIMT.
The Authors have presented sufficient data. The appropriate tables and figures have been provided. The
article is easy to read and logically structured. The methods are adequately described. The authors used
appropriate statistical methods. The conclusions are consistent with the presented evidence and
arguments.
1. Please add the inter- and within-test coefficients of variation (CV) of Angiogenin
Answer: We kindly thank the Reviewer for a positive opinion about our study. For the whole study group the
coefficients of variation (CV) for angiogenin was 34.6%. For the healthy controls and DM1 patients, CV was
20.52% and 41.6%, respectively. Parameters for the coefficient of variation of angiogenin serum levels have
been added to the table 3.
2. Please add more limitations to your study: In future studies, it will be worth estimating the
correlations between angiogenin and endothelial dysfunction estimated by flow-mediated dilation.
Brachial flow-mediated dilation (FMD) is a physiologic measure and Carotid IMT is an anatomic
structural measure of subclinical atherosclerosis [PMID: 36141513; PMID: 27765534]; too small groups,
etc.
In future studies, it will be worth estimating the correlations between angiogenin and endothelial
dysfunction estimated by flow-mediated dilation. Brachial flow-mediated dilation (FMD) is a physiologic
measure and Carotid IMT is an anatomic structural measure of subclinical atherosclerosis
Answer: The material analyzed is the result of the implementation of a protocol that includes that includes the
assessment of both micro- and microcirculation. We are fully aware of the fact that not all interesting methods
have been applied. We share the opinion of the Reviewer that the FMD would make a valuable contribution to
our research.
Therefore, in the revised version of our manuscript, we have added a new paragraph to the discussion section:
“The limitation of the present study is the relatively small number of subjects. The imbalance between the
number of participant in each subgroup may be a limitation of our study, which could potentially influence the
power of study. Our results require further studies with larger cohorts of patients to validate the results obtained.
Another limitation of our study was the use of cIMT as the sole marker of atherosclerosis and vascular function.
The use of additional methods would provide valuable additional information. For instance, FMD could help
understand endothelial dysfunction in diabetic patients with metabolic syndrome. This demonstrates the potential
for future studies in this area [59].”
3. In the introduction section, please add more information about metabolic syndrome and its influence
on CV risk.
Answer:
Therefore, in the revised version of our manuscript, we have added a new paragraph to the Introduction section.
The term metabolic syndrome (MS) was introduced to emphasize the cumulative effect of several risk factors.
Various studies have shown an increased risk of CVD in MS, regardless of the MS criteria used [22-26]. Mottillo
found a 2-fold increased risk of CVD, CVD mortality and stroke and a 1.5-fold increased risk of mortality from
any cause associated with MS in the largest systematic review and meta-analysis (n = 951,083) [27]. However,
more recent studies have shown that the associations between MS and CVD outcomes are either null or weak
[28, 29]. The RIVANA [30] study, conducted in Mediterranean populations, found that MS was independently
associated with incident CVD, CVD-related mortality and all-cause mortality. However, its association with
myocardial infarction or stroke could not be established. Each MS component was also independently associated
with all study outcomes, with similar strength to MS alone. This study showed that the risk of cardiovascular
events, with the exception of myocardial infarction, increased together with the increasing number of MS
components. However, there may be differences in the risk of cardiovascular events depending on the
combination of individual criteria.
Either defined by the ATP III or the IDF criteria, metabolic syndrome consists of five metabolic abnormalities
and indicates a substantial lifetime risk of cardiovascular disease. However, it is not a reliable indicator of 10
year cardiovascular risk. A global risk assessment, such as the Framingham score, is still necessary for the
determination of 10-year risk [31].
We believe that central obesity and its metabolic consequences play a substantial role in the progression
of type 2 diabetes and hypertension, as well as the exacerbation of preexisting conditions. For instance,
overweight patients with type 1 diabetes experience hyperinsulinemia and insulin resistance, leading to the
development of double diabetes. Merger et al [32] performed a study of 31119 people with type 1 diabetes,
searching for people with MS and analyzing them for late vascular complications. The study revealed that 25.5%
of this population had both T1DM and MS. Furthermore, those with dual diabetes showed an increased
prevalence of macrovascular comorbidities including coronary heart disease (8.0% compared to 3.0% without
MS), stroke (3.6% compared to 1.6%), and diabetic foot syndrome (5.5% compared to 2.1%). Microangiopathic
complications were found to be twice as frequent than in the group without MS (retinopathy 32.4% versus
21.7%, nephropathy 28.3% versus 17.8%). Vascular complications were more common in the double diabetes
group, regardless of glucose control. The team therefore concluded that it is important to identify patients at risk
of double diabetes and prevent them from developing MS.

Reviewer 3 Report
This study investigated the relationship between angiogenin levels and carotid intima-media thickness in type 1 diabetes mellitus patients with and without the metabolic syndrome. Overall, the study reports negative findings. 1. I do not understand why the IDF criteria for metabolic syndrome were used. For a discussion of the importance of specific criteria of metabolic syndrome, see e.g. PMID: 37158487. 2. This is a cross-sectional study. It is essential to know how controls were chosen. 3. What was the original research hypothesis? When reading the abstract, the reader gets the impression that multiple post-hoc analyses were performed in order to find a significant difference. Several categories were excluded. This seems to be data dredging. Overall, I would conclude that is a negative study. There is nothing wrong with that in itself. 4. Carotid IMT is a very poor surrogate marker of coronary atherosclerosis. The endpoint itself is dubious. 5. The first paragraph of the conclusion is not to the point. I appreciate the final paragraph of the conclusion, which clearly confirms that the study is negative.
Author Response
Reviewer #3:
This study investigated the relationship between angiogenin levels and carotid intima-media thickness in
type 1 diabetes mellitus patients with and without the metabolic syndrome. Overall, the study reports
negative findings.
1. I do not understand why the IDF criteria for metabolic syndrome were used. For a discussion of the
importance of specific criteria of metabolic syndrome, see e.g. PMID: 37158487.
Answer:
We added a paragraph to the discussion section.
“In 2009, researchers agreed that abdominal obesity should not be a prerequisite for the diagnosis of metabolic
syndrome, but rather only one of the 5 criteria. Currently, the presence of 3 out of the 5 criteria is required for
diagnosis [38].
It should be mentioned that although metabolic syndrome is predominantly identified in overweight individuals,
it can sometimes occur in lean people. An important finding from the Osadnik study [39] was that the presence
of MS in lean patients carries the same adverse long-term prognosis as in obese patients.
In our study we used the IDF criteria, which identified patients with numerous car-diovascular risk factors
beyond type 1 diabetes. Patients with MS met the waist criterion, 4 of them met the BMI criterion, and there
were some who did not have MS despite central obesity.
We did not use the ATP III criteria in our study as we felt they were not sufficiently restrictive. According to the
IDF, 35.7 % diabetic patients had MS, while ac-cording to the ATP III just 28.6%. The new criteria proposed by
the Polish researchers are based on the presence of central obesity (waist 88/102 cm or BMI> 30 kg/m2),
carbohydrate disorders , hypertension and lipid disorders expressed as non-HDL cholesterol levels or
hypolipemic treatment. Only 12.5% of our patients met Polish criteria [40].”
2. This is a cross-sectional study. It is essential to know how controls were chosen.
Answer: The control group was recruited from coworkers and their families.
We added this information to the article.
3. What was the original research hypothesis? When reading the abstract, the reader gets the impression
that multiple post-hoc analyses were performed in order to find a significant difference. Several categories
were excluded. This seems to be data dredging. Overall, I would conclude that is a negative study. There is
nothing wrong with that in itself.
Answer: In the last few years we have focused our interest on relationship between cytokines and vascular
dysfunction in patients with type 1 diabetes. Metabolic syndrome refers to a cluster of risk factors widely
accepted for identifying patients at high cardiovascular risk. Metabolic syndrome occurs in about 34% of
patients with type 1 diabetes. We thought that it will be of interest to describe the impact of metabolic syndrome
on the relationship between angiogenin and cIMT. We believe we have selected our study groups in a way to
minimize any confounding factors, hence exclusions.
4. Carotid IMT is a very poor surrogate marker of coronary atherosclerosis. The endpoint itself is
dubious.
Answer: There are still many controversies about the value of cIMT as a prognostic tool of cardiovascular
risk, but to clarify the issue we added a paragraph to the Introduction: we hope that by adding these
paragraphs we put ourselves in line with the revised point of view.
“Recent results from a meta-analysis carried out by Meimei et al in August 2023 suggest that there is an
association between carotid ultrasound parameters and the microvascular and macrovascular complications
of diabetes. Therefore, Meimei et al proposed that in patients with diabetes, carotid ultrasound may be a
valuable tool for predicting patient outcomes [14]. Abnormalities in parameters such as carotid intimacy
media thickness , pulse wave velocity [15] and central amplification index can reflect potential vascular pathology. The researchers [16] found that there was an independent association between CIMT and cerebral
microbleeds in young individuals with type 1 diabetes despite no clinical signs of neurological disease,
indicating a link between subclinical systemic atherosclerosis and early microvascular pathology in the brain.
In a meta-analysis, Wang et al [17] showed that cIMT, endothelium-dependent flow-mediated dilation
(FMD), carot-id-femoral pulse wave velocity, and glyceryl trinitrate-mediated dilatation are significant-ly
different between patients with type 1 diabetes and control patients. Their results are in line with the current
evidence for increased CV risk in type 1 diabetic patients, and stress the benefit in identifying markers of
subclinical atherosclerosis in their treatment. Another review indicated that arm FMD is lower and cIMT is
higher in obese adults. Consequently, changes in arm FMD and cIMT may be important surrogate markers
appropriate for use in clinical trials [18]. It should be noted that the cIMT is more widely used in everyday
clinical practice. The Epidemiology of Diabetes Interventions and Complications (EDIC) or Diabetes Control
and Complications Trial (DCCT) found an association between cIMT and subsequent coronary events in
patients with type 1 diabetes. However, this association was no longer found after adjusting for traditional
cardiovascular risk factors [19].
Atherosclerosis progresses over a number of years in the absence of symptom. Therefore, identifying
patients who are at risk for developing symptomatic disease offers avenues for primary prevention. Previous
studies have shown that various interventions reduce cIMT progression are associated with a significant
reduction in CVD (cardiovascular disease) risk, supporting its use as a surrogate marker. These findings
further sup-port the use of cIMT as a marker of increased risk for atherosclerosis, potentially helping identify
individuals at risk for early development of advanced atherosclerosis [20].“
5. The first paragraph of the conclusion is not to the point. I appreciate the final paragraph of the
conclusion, which clearly confirms that the study is negative.
Answer:
Based on the Reviewer's comments, we are clarifying the conclusions section and would like to thank the
Reviewer for the valuable comments.
A clear hypothesis has been added to the conclusion of the introduction.
“Our report is dedicated to the hypothesis that the presence of metabolic syndrome is associated with increased
angiogenin concentration and cIMT”.

Reviewer 4 Report
The manuscript submitted by Neubauer-Geryk et al., titled: "Angiogenin levels and carotid intima-media thickness in patients with type 1 diabetes and metabolic syndrome" is an interesting study investigating the relationship between levels of angiogenic and carotid intima-media thickness in T1DM patients. This is an interesting topic with potentially important clinical applications.
The reviewer would like to raise the following points for the improvement of the manuscript.
1. The introduction seems rather short. There are also statements that are not referenced (such as the opening one).
2. It would be helpful to include more statistics (prevalence and burden of disease) and clinical information in the introduction, referenced.
3. It would be helpful to have the hypothesis of the work stated at the end of the introduction.
4. How was the number of participants in the study determined (power calculation)?
5. Consider specifying more directly/clearly articulated the inclusion and exclusion criteria for the study.
6. Were dietary intakes considered in the analyses? Calories, carbohydrates, timing of meals, (sodium intake?). All these can extend significant effects both to T1DM outcomes and potentially, blood pressure and carotid thickness.
7. How did the authors control for confounding factors?
English needs some polishing from an English native speaker preferably.
Author Response
Reviewer #4:
The manuscript submitted by Neubauer-Geryk et al., titled: "Angiogenin levels and carotid intima-media
thickness in patients with type 1 diabetes and metabolic syndrome" is an interesting study investigating
the relationship between levels of angiogenic and carotid intima-media thickness in T1DM patients. This
is an interesting topic with potentially important clinical applications.
Thank you for the positive opinion about our report.
The Reviewer would like to raise the following points for the improvement of the manuscript.
1. The introduction seems rather short. There are also statements that are not referenced (such as the
opening one).
Answer: In response to the reviewers comment, we have expanded the introduction and added all necessary
references.
2. It would be helpful to include more statistics (prevalence and burden of disease) and clinical
information in the introduction, referenced.
Answer: We hope that the point raised by the Reviewer is covered by the extended content of the introduction:
“Although mortality risk related to type 1 diabetes has fallen by 29% over the past 10 years, it is still estimated
to be between two to eight times higher than in the general population [1, 2]. Thus, ESC guidelines recommend
aggressive treatment of cardiovascular risk factors in patients with type 1 diabetes, particularly those over the
age of 40 or with evidence of microvascular complications [3]. In 2021, there will be about 8.4 million people
with diabetes, including 18% of patients under the age of 20, 64% aged 20-59, 19% aged 60 and over. It is
estimated that one million new cases will be diagnosed in 2023, with the median age of onset of 39 years.
Remaining life expectancy for a 10-year-old diagnosed with type 1 diabetes in 2021 ranges from 13 years in
low-income countries to 65 years in high-income countries [1].”
3. It would be helpful to have the hypothesis of the work stated at the end of the introduction.
Answer: A clear hypothesis has been added to the conclusion of the introduction.
“Our report is dedicated to the hypothesis that the presence of metabolic syndrome is associated with increased
angiogenin concentration and cIMT”.
4. How was the number of participants in the study determined (power calculation)?
Answer:
Our data come from the pilot phase of an extended study on the development of micro- and macroangiopathy in
adolescents with type 1 DM.
We add the information about power of the test at the end of Result section.
Calculated Power of the test yielded 0.828.
5. Consider specifying more directly/clearly articulated the inclusion and exclusion criteria for the study.
Answer: We have expanded the criteria used to include and exclude patients.
“Informed consent was the principal consideration. The study did not include diabetic patients who met the
exclusion criteria: renal insufficiency, uncontrolled diabetes or ketoacidosis, chronic respiratory disease, history
of cardiovascular events, history of clinical manifestations of cerebral circulation disorders and/or with focal
symptoms of central nervous system damage confirmed by neurological examination. In addition, patients with
severe hypoglycaemia within 30 days prior to the examination were excluded from the cohort. The factor that
excluded healthy people or people with diabetes from the study was the presence of systemic diseases such as
rheumatoid arthritis or psoriasis. People were included if they had a diagnosed medical history of both
hypothyroidism and hyperthyroidism and were euthyroid at the time they qualified for the study. Euthyroidism
was confirmed by testing TSH and fT4 hormone levels in the month prior to the study. In the absence of
euthyroidism, these tests were performed as part of the biochemical tests. Pregnancy and alcohol abuse were also
exclusion criteria in both groups”.
A detailed statement of inclusion and exclusion criteria is presented in the group description.
6. Were dietary intakes considered in the analyses? Calories, carbohydrates, timing of meals, (sodium
intake?). All these can extend significant effects both to T1DM outcomes and potentially, blood pressure
and carotid thickness.
Answer: The patients remained on either a standard diet (control group) or a standard diabetic diet (patients with
T1DM). Prior to data collection, the subjects did not report any significant changes in weight
7. How did the authors control for confounding factors?
Answer:
Strict adherence to the inclusion and exclusion criteria was the most important. To maintain an appropriate blood
glucose range during testing, blood glucose was checked by the patient using their own blood glucose meter.
Body temperature was also closely monitored. A stable temperature was maintained during the examinations.
The insulin dosage per kg was comparable between subgroups of type 1 DM. Thus we add this information to
the table 1.
Comments on the Quality of English Language
English needs some polishing from an English native speaker preferably.
Answer: The current version of the manuscript with the corrections based on the referees' comments will be
checked again by one of the authors (MW), who is a medicine doctor affiliated with Department of Pediatrics,
Northwestern University Feinberg School of Medicine; Division of Neonatology, Ann & Robert H. Lurie
Children's Hospital of Chicago.

Round 2
Reviewer 3 Report
The authors have provided a sufficient answer to the comments that I raised. Of course, certain concerns that were valid remain but the revised version has been sufficiently improved.
Author Response
Thank you very much for the acceptance of the changes that have been made to the manuscript.
Reviewer 4 Report
The authors have made a reasonable effort in addressing reviewer's comments. Proofreading is suggested.
Author Response
We agree with the Reviewer's comments.
We have included a figure (Result paragraph) that shows the levels of IMT and angiogenin in subgroups of patients categorized according to the status of the metabolic syndrome. However, we are concerned that further expansion of the Introduction may not be accepted by the other Reviewers, as this part of the manuscript is currently very extensive.
We have once again reviewed the text and made a few linguistic correction. We would like to emphasize, however, that the manuscript was co-written and edited by an English native speaker, Melanie Wielicka.
We hope the Reviewer will be satisfied with the current revised manuscript.